# Changes in attitudes to vaccination as a result of the COVID-19 pandemic: A longitudinal study of older adults in the UK

**Allyson J. Gallant**[1], **Louise A. Brown Nicholls**[2], **Susan Rasmussen**[2], **Nicola Cogan**[2], **David Young**[3], **Lynn Williams** [2]*

**1** Faculty of Health, Dalhousie University, Halifax, Nova Scotia, Canada, **2** School of Psychological Sciences & Health, University of Strathclyde, Glasgow, Scotland, United Kingdom, **3** Department of Mathematics & Statistics, University of Strathclyde, Glasgow, Scotland, United Kingdom

* lynn.williams@strath.ac.uk

**Data Availability Statement:** The quantitative data that support the findings of this study are openly available via Open Science Framework at https://doi.org/10.17605/OSF.IO/JYPN9.

## Abstract

### Background

The rapid development of COVID-19 vaccines has brought an unprecedented focus on public attitudes to vaccines, with intention to accept a COVID-19 vaccine fluctuating during the pandemic. However, it is unclear how the pandemic may influence attitudes and behaviour in relation to vaccines in general. The aim of the current study is to examine older adults' changes in vaccination attitudes and behaviour over the first year of the pandemic.

### Methods

In February-March 2020 (before the first COVID-19 national lockdown in the UK), 372 older adults (aged 65+) provided sociodemographic information, self-reported influenza vaccine uptake, and completed two measures of vaccination attitudes: the 5C scale and the Vaccination Attitudes Examination Scale. One-year later, following rollout of COVID-19 vaccines to older adults, participants provided information on their COVID-19 and influenza vaccine uptake in the previous 12 months, and completed the 5C and VAX scales again. Paired samples t-tests were used to examine changes in vaccination attitudes over time.

### Results

Almost all participants (98.7%) had received at least one dose of a COVID-19 vaccine, and a significant increase in influenza uptake was identified (83.6% in 2020 to 91.6% in 2021). Complacency, mistrust of vaccine benefit, concerns about commercial profiteering, and constraints to vaccination had significantly decreased between Time 1 and Time 2, and collective responsibility had significant increased. However, calculation and worries about unforeseen future effects had increased, indicating that participants now perceived higher risks related to vaccination and were taking a more deliberative information-seeking approach.

**Funding:** The study was funded by the Chief Scientist Office https://www.cso.scot.nhs.uk/, Scottish Government (Ref: Ref: CGA/19/52) research grant awarded to LW, LBN, SR, NC. The funders had no role in study design, data collection and analysis, decision to publish, or preparation of the manuscript.

**Competing interests:** The authors have declared that no competing interests exist.

**Abbreviations:** RQ, Research Questions; VAX, Vaccination Attitudes Examination.

## Conclusion

The results show significant changes in vaccination attitudes across the pandemic. These changes suggest that while older adults became less complacent about the importance of vaccines, concerns about potential risks associated with vaccination increased. It will be important for public health communication to address these concerns for all vaccines offered to this group.

## Introduction

Since COVID-19 was first identified as a pandemic by the World Health Organisation (WHO) in March 2020 [1], there has been worldwide attention placed on controlling the spread of the virus and the development of effective vaccines. Vaccination against COVID-19 has also been highlighted as a critical component of the recovery strategy for the UK to return to normal [2]. As multiple safe and effective vaccines against COVID-19 have been developed and approved for use, factors such as varying efficacy rates across these COVID-19 vaccines [3], effectiveness against newly identified variants of concern [4], and a reported link of the AstraZeneca vaccine to the rare development of blood clots [5] have only intensified concerns about vaccinating against COVID-19.

In most countries, including the UK, older adults have been identified as a priority group for receiving COVID-19 vaccines [6]. In addition, older adults are also routinely offered vaccines against seasonal influenza, pneumococcal disease, and shingles. Due to older adults' increased risk of acquiring these illnesses, and their increased likelihood of developing complications as a result, high levels of vaccination uptake in this age group are particularly important [7]. However, acceptance rates across these vaccines vary. Uptake of the annual influenza vaccine among older adults in the UK is consistently around 70%, reaching over 80% uptake during the 2020–2021 influenza season, likely as a result of intensified vaccination campaigns due to COVID-19 [8]. In contrast, uptake rates of the pneumococcal vaccine is 69% in adults 65 and older, and shingles vaccine uptake is only 54% among adults aged 70–79, with only slight increases in uptake rates for both vaccines seen in recent years [9, 10]. Commonly identified reasons for poor uptake of these vaccines include low levels of confidence and not prioritizing vaccinations [11, 12]. Given up to 90% of the population will need to vaccinate against COVID-19 to achieve herd immunity [13], it is critical to understand vaccination views among this priority population and how views may have changed since the beginning of the pandemic.

There are a number of factors that influence older adults' vaccine hesitancy, or "the unwillingness or refusal to vaccinate despite accessible vaccine services" [14]. The 3C model developed by MacDonald & the SAGE Working Group on Vaccine Hesitancy was designed to capture the nuances amongst factors affecting vaccine hesitancy. These factors include Confidence (e.g., trust in the safety and effectiveness of vaccines, vaccine developers and government), Complacency (e.g., vaccination is a low priority or vaccine-preventable diseases are not a concern) and Convenience (e.g., time or geographical factors, quality of accessible vaccines) [14]. Furthermore, levels of hesitancy can vary depending on the vaccine. A variety of tools and instruments have been designed to investigate psychosocial aspects of vaccine hesitancy [15–18]. However, to date few of these instruments have been applied in older adult populations.

Among factors affecting older adults' vaccine uptake, a low sense of collective responsibility, preference for natural immunity, distrust of commercial vaccination companies and high

levels of information seeking have been identified as predictors of vaccine hesitancy [19]. However, while a sense of collective responsibility has been identified as a predictor of older adults' uptake of a range of vaccines (i.e. influenza, pneumococcal, and shingles vaccines), concerns about profiteering predicted lack of uptake of the pneumococcal and shingles vaccines, whose uptake was considerably lower than the influenza vaccine [19]. Overall, this suggests that understanding of disease risk and vaccine benefits, along with the broader psychosocial factors specifically associated with uptake, vary by vaccine. While high levels of intended uptake of a COVID-19 vaccine have been identified in older adult populations, worries of potential vaccine side effects and rapid vaccine development have been identified as notable concerns which may affect vaccine acceptance and uptake [20].

Across the span of the pandemic, intention to vaccinate against COVID-19 has fluctuated [21], with recent findings from the United States identifying a decline in intentions to vaccinate against COVID-19 over time [22]. However, it is not known what influence the pandemic has had on older adults' attitudes and behaviour in relation to vaccines. The attitudes and behaviours of older adults are of particularly importance given their relative vulnerability to COVID-19 and other vaccine-preventable disease, and their priority status to receive a COVID-19 vaccination in the UK [23]. While the pandemic has highlighted the importance of vaccination, the focus on safety concerns and potential side effects, and spread of misinformation relating to the COVID-19 vaccines may have made people less confident about vaccines in general [24]. There is a need to identify the impact of the COVID-19 pandemic on vaccination attitudes and behaviours towards vaccinations in order to inform future communications around vaccines. Therefore, we aimed to address three research questions (RQ's):

RQ1: How have older adults' (65+) vaccination attitudes changed over the previous 12 months?

RQ2: What were the key factors that shaped older adults' decision to receive a COVID-19 vaccine, in terms of motivating factors and concerns?

RQ3: Has older adults' influenza vaccine uptake changed over the previous 12 months and what are their vaccination intentions for the next 12 months?

## Methods

### Participants and procedure

The present study consisted of an online, longitudinal survey over two time points. The first survey (Time 1) was conducted in February-March 2020. During this time, cases of COVID-19 had started to be identified in the UK, but data collection concluded prior to the introduction of any lockdown restrictions. The second survey (Time 2) was conducted one year later, in March 2021. At this time, all adults in the UK aged 65 and over had been offered the first dose of a COVID-19 vaccine. The target population for the survey was therefore older adults (aged 65 and older) living in the UK. Participants were eligible to take part if they were living independently in the community (i.e., not living in a care facility) and were generally in good health (and, specifically, not diagnosed with a neurological condition). At Time 1, participants were recruited through emails to university participation panels and social media posts on Facebook and Twitter, inviting them to complete an online survey via Qualtrics. At Time 2, we re-contacted participants via email and invited them to complete the follow-up survey, also administered through Qualtrics. All materials and procedures were approved by the School of Psychological Sciences and Health, University of Strathclyde Ethics Committee and all participants gave informed consent.

At Time 1, participants provided sociodemographic information and self-reported influenza vaccine uptake in the previous 12 months. Participants also completed the 5C scale [16] and the Vaccination Attitudes Examination Scale (VAX) [25]. At Time 2, participants provided information on their COVID-19 and influenza vaccination uptake in the previous 12 months and completed the 5C and VAX scales again. In addition, participants were also asked about the factors that influenced their decision to vaccinate against COVID-19, if they had any concerns about the COVID-19 vaccines, and the sources of information they consulted in order to inform their COVID-19 vaccination decision. Finally, participants were asked about their intention to receive the annual influenza vaccine and any booster doses of a COVID-19 vaccine in the future.

## Measures

**5C scale.** The 5C scale [16] assesses the psychological antecedents to vaccination. It is comprised of five, three-item subscales to measure: Confidence (e.g., *'I am confident that public authorities decide in the best interest of the community'*), Complacency (e.g., *'vaccination is unnecessary because vaccine-preventable diseases are not common anymore'*), Constraints (e.g., *'it is inconvenient to receive vaccinations'*), Calculation (e.g., *'for each and every vaccination, I closely consider whether it is useful for me'*), and Collective Responsibility (e.g., *'vaccination is a collective action to prevent the spread of diseases'*). Responses were measured on a seven-point Likert scale (1 = strongly disagree, 7 = strongly agree) and scored by calculating the mean score for each subscale (score range 1–7). Higher confidence and collective responsibility scores indicate enablers to vaccination, while higher complacency, calculation, and constraints scores indicate more individual barriers to vaccination [16, 26].

**VAX scale.** The VAX scale [25] consists of 12 items to identify individuals' attitudes towards vaccination. The scale contains four, three-item subscales: Mistrust of Vaccine Benefit (e.g., *'I can rely on vaccines to stop serious infectious diseases'*), Worries about Unforeseen Future Effects (e.g., *'I worry about the unknown effects of vaccines in the future'*), Concerns about Commercial Profiteering (e.g., *'authorities promote vaccination for financial gain, not for people's health'*), and Preference for Natural Immunity (e.g., *'natural exposure to viruses and germs gives the safest protection'*). All items were measured on a five-point scale (1 = strongly disagree, 5 = strongly agree) and were scored by calculating the mean scores for each subscale as well as a mean total score (range 1–5). Lower scores indicate more positive vaccination views while higher scores represent more negative views [25].

**COVID-19 vaccination questions.** Participants were asked to rank-order the factors which influenced their decision to vaccinate, with the response options of: "to protect personal health"; "to protect the health of family and friends"; "to achieve herd immunity"; and "to end restrictions". A free-text option was provided to capture any additional factors. Participants were also asked if they had any concerns about the COVID-19 vaccines, including: concerns about side effects; safety concerns; speed of vaccine development; and concern the vaccine is not effective. A free-text option was included again so that participants could detail any other concerns. Finally, two questions asked about intention to receive influenza and COVID-19 vaccinations in future (e.g., "If you were offered further vaccinations against COVID-19 would you accept them?") with the response options of 'yes', 'no', and 'not sure'. A copy of the survey can be found in S1 File.

## Data analysis

Paired samples t-tests were used to test for differences in scores on the subscales of the 5C and VAX measures across the two time-points. In addition, descriptive statistics were produced for

influenza and COVID-19 vaccine uptake, along with the main factors that shaped the participants' COVID-19 vaccination decision-making. Cohen's *d* was calculated in each case as an estimates of the effect sizes of time. Effect sizes of 0.2 were interpreted qualitatively as a small effect, 0.5 were interpreted as medium effect and 0.8 were interpreted as a large effect [27]. Statistical analyses were conducted using IBM SPSS (version 26) at 5% significance levels.

## Results

### Characteristics of participants

The survey at Time 1 was completed by 372 participants, and 311 of these participants completed the follow-up survey at Time 2, representing an 83.6% follow-up rate. Participant characteristics at both time points are shown in Table 1. We compared the sociodemographic characteristics of those who participated at both time points, with those who completed only Time 1. There were no significant differences between the completers and non-completers based on age, gender, or deprivation level. Almost all participants (n = 307; 98.7%) indicated that they had received at least one dose of a COVID-19 vaccine.

### RQ1: How have older adults' vaccination attitudes changed over the previous 12 months?

For the 5C subscales, paired samples t-tests indicated that the four subscales of complacency, constraints, calculation, and collective responsibility had changed significantly across the two time points. However, there was no significant change in vaccine confidence (see Table 2). Both complacency and constraints to vaccination had significantly decreased at Time 2 indicating that participants had become less complacent about vaccination and perceived fewer constraints to accessing vaccination. In addition, scores on the collective responsibility

**Table 1. Sociodemographic and vaccine uptake information for the sample at Time 1 and Time 2.**

| Variables | Time 1 (n = 372) | | Time 2 (n = 311) | |
|---|---|---|---|---|
| | **n** | **%** | **n** | **%** |
| **Age** | M = 70.5 (SD = 4.6) | | M = 70.3 (SD = 4.7) | |
| **Gender** | | | | |
| Male | 184 | 49.7% | 159 | 51.5% |
| Female | 184 | 49.7% | 149 | 48.2% |
| Prefer not to say | 2 | 0.5% | 1 | 0.3% |
| **Deprivation Quintile** | | | | |
| 1 (most deprived) | 57 | 16.3% | 46 | 15.5% |
| 2 | 83 | 23.8% | 76 | 25.7% |
| 3 | 69 | 19.8% | 55 | 18.6% |
| 4 | 91 | 17.5% | 53 | 17.9% |
| 5 (least deprived) | 79 | 22.6% | 66 | 22.3% |
| **Received annual influenza vaccination** | 311 | 83.6% | 285 | 91.6% |
| **COVID-19 vaccination** | | | | |
| First dose received | - | - | 302 | 97.1% |
| Both doses received | - | - | 5 | 1.6% |
| Declined | - | - | 3 | 1% |
| Not yet been offered | - | - | 1 | 0.3% |

*Note. M* = mean; *SD* = standard deviation.

**Table 2. Means and Cronbach's alphas for 5C and VAX-measured vaccination attitudes over 12 months.**

| Scale | Subscale | Time 1 | | Time 2 | | *p*-value | *Cohen's d* |
|---|---|---|---|---|---|---|---|
| | | *M (SD)* | Cronbach's α | *M (SD)* | Cronbach's α | | |
| 5C | Confidence | 6.14 (1.34) | 0.87 | 6.20 (1.25) | 0.83 | 0.53 | - |
| | Complacency | 2.00 (1.10) | 0.49 | 1.65 (0.99) | 0.57 | < .01 | 0.300 |
| | Constraints | 1.31 (0.72) | 0.68 | 1.19 (0.59) | 0.52 | < .01 | 0.171 |
| | Calculation | 5.24 (1.71) | 0.79 | 5.49 (1.57) | 0.80 | 0.02 | 0.137 |
| | Collective Responsibility | 6.23 (1.06) | 0.53 | 6.43 (1.03) | 0.63 | < .01 | 0.172 |
| VAX | Mistrust of Vaccine Benefits | 1.98 (0.75) | 0.87 | 1.88 (0.65) | 0.84 | 0.02 | 0.134 |
| | Worries About Unforeseen Future Effects | 2.74 (0.79) | 0.74 | 2.87 (0.72) | 0.70 | < .01 | 0.196 |
| | Concerns About Commercial Profiteering | 1.65 (0.74) | 0.83 | 1.56 (0.65) | 0.78 | 0.02 | 0.186 |
| | Preference for Natural Immunity | 2.12 (0.85) | 0.83 | 2.11 (0.81) | 0.82 | 0.91 | - |

*Note. M* = mean; *SD* = standard deviation; VAX = Vaccination Attitudes Examination.

subscale had increased indicating that participants felt an increased sense of collective responsibility to vaccinate at Time 2. However, calculation scores had increased, indicating that participants now perceived higher risks related to vaccination and were taking a more deliberative information seeking approach to inform their vaccination decision making. The effect sizes for these changes were generally small (Cohen's *d* = 0.13–0.30) with the largest effect being observed for complacency (*d* = 0.30).

Across the VAX scale, there were significant differences between time points for the three subscales of mistrust of vaccine benefit, worries about unforeseen future effects, and concerns about commercial profiteering. However, there was no significant difference relating to the preference for natural immunity subscale across the time points. Mistrust of vaccine benefits and concerns about commercial profiteering both significantly decreased between Time 1 and Time 2. However, worries about unforeseen future effects had significantly increased from Time 1 to Time 2. The effect sizes for these significant changes were also small (Cohen's *d* = 0.13–0.20).

### RQ2: What were the key factors that shaped older adults' decision to receive a COVID-19 vaccine in terms of motivating factors and concerns?

When asked to rank-order the importance of factors affecting the decision to vaccinate against COVID-19, protecting personal health was ranked by the majority of participants as the most important factor (n = 162; 52%), followed by protecting the health of friends and family (n = 43; 14%), contributing to herd immunity (n = 34; 11%) and to bring an end to the pandemic restrictions (n = 7; 2%; see Table 3). Additional free-text responses included: feeling it was a sensible decision or brought peace of mind (n = 7; 2%); to protect the health of

**Table 3. Rank-ordering of factors affecting COVID-19 vaccine uptake.**

| | Ranked Importance of Factor | | | |
|---|---|---|---|---|
| | N (%) | | | |
| | First | Second | Third | Fourth |
| **To protect myself** | 162 (52%) | 52 (17%) | 25 (8%) | 8 (3%) |
| **To protect family and friends** | 43 (14%) | 162 (52%) | 33 (11%) | 9 (3%) |
| **To help achieve herd immunity** | 34 (11%) | 20 (6%) | 149 (48%) | 41 (13%) |
| **To bring an end to restrictions** | 7 (2%) | 13 (4%) | 37 (12%) | 184 (59%) |

vulnerable community members (n = 4; 1.3%); to be able to travel (n = 4; 1.3%); and to support a return to normalcy (n = 2; 0.6%).

When asked if participants had any concerns about the COVID-19 vaccines, 72% (n = 225) indicated that they had no concerns. Among those who identified concerns, worries of potential vaccine side effects (n = 49; 15.8%) and vaccine efficacy concerns (n = 46; 14.8%) were the most common responses. Worries about the speed of vaccine development (n = 21; 6.8%) and vaccine safety fears (n = 9; 2.9%) were also identified. Free-text responses to this item also identified that some participants (n = 3; 1%) were concerned about the time window between the first and second doses, which was an average of 12 weeks in the UK at the time of survey.

### RQ3: How has older adults' influenza vaccine uptake changed over the previous 12 months and what are their vaccination intentions for the next 12 months?

As shown in Table 1, 83.6% (n = 311) of participants received the annual influenza vaccination in the 12 months prior to Time 1. This rose to 91.6% (n = 285) in the 12 months prior to Time 2, representing a significant increase in influenza vaccination across the time points ($p < .01$). In terms of future intention to vaccinate at Time 2, 91.6% (n = 285) intend to receive the annual influenza vaccine in the next influenza season, and 97.1% (n = 302) intend to accept any future COVID-19 booster vaccines they are offered.

## Discussion

To our knowledge, the current study is the first to assess longitudinal changes in general vaccination attitudes and behaviours among older adults during the first year of the COVID-19 pandemic. Our findings suggest that older adults' attitudes towards vaccination have changed significantly as a result of the pandemic. Several aspects have been positively influenced, with complacency, mistrust of vaccine benefit, concerns about commercial profiteering, and perceived constraints to vaccination significantly decreasing between Time 1 and Time 2. Sense of collective responsibility significantly increased. These results suggest that the pandemic has led people to re-evaluate the importance of vaccination and increase their willingness to protect others. In addition, the decrease in perceived constraints to vaccination suggests that people now perceive fewer structural and psychological barriers to vaccination, suggesting that the innovations in service delivery of COVID-19 vaccines has been effective in reducing some practical barriers [6].

While the above results would suggest that attitudes to vaccination have generally become more positive over the first year of the pandemic, we also found that calculation had increased. This may indicate that participants now perceived higher risks related to vaccination and were taking a more deliberative, information-seeking approach to inform their vaccination decision-making. A recent evidence synthesis of factors affecting COVID-19 vaccination identified use of traditional media sources (e.g., local and national televised news programmes, newspapers and radio formats), government and public health information sources were associated with improved COVID-19 vaccine acceptance [28]. With increases in information-seeking among older adults, it is crucial to ensure high-quality and trusted information sources are sought out and easily accessible to this population to support their vaccination decisions.

We identified that concerns about vaccine side effects have significantly increased during the pandemic. The unprecedented focus on the vaccine approval process, and widespread coverage of side effects of the vaccines, particularly evident for the AstraZeneca vaccine [29], seems to have led older adults to be more concerned about the side effects of vaccines in general, or at least to seek and evaluate safety-related information. These findings align with

previous research relating to COVID-19 vaccines, where participants have raised persistent concerns about vaccine side effects [30–32]. Our results suggest that it will be important to consider how to address these concerns, not only for the COVID-19 vaccines but also for others. The increase in calculation scores also suggest that older adults may now seek out additional information about any vaccines they are offered. Consequently, there is a need to ensure that we have educational interventions and resources in place which will provide credible, trusted and transparent information about vaccinations and potential side effects.

Protecting personal health was identified as a key factor which shaped the decision to vaccinate against COVID-19, followed by protecting friends and family, contributing towards herd immunity, and ending the pandemic-related restrictions. These findings align with previous research that identified personal health, the health consequences to others, and peace of mind as main facilitators to future intended COVID-19 vaccination [20, 33]. Considering up to 90% vaccine uptake may be required to achieve herd immunity against COVID-19 [13], it is worth highlighting that vaccine uptake among this priority population has been driven by multiple personal and social factors. In those that remain hesitant towards vaccination against COVID-19, it may be beneficial to emphasise the broader societal and environmental benefits of vaccination in addition to the personal health benefits, bearing in mind specific disease risks [19].

In relation to vaccination behaviour, we identified near universal uptake of at least one dose of a COVID-19 vaccine. While the uptake rate observed in our sample is very high, it is in line with the COVID-19 vaccination rates among older adult populations across Scotland and the UK [34, 35]. We also found significantly higher influenza vaccination rates for the 2020–2021 influenza season compared to that in 2019–2020. It is important to note these identified high uptake rates of COVID-19 and influenza vaccines may be context-dependent, and there are a number of broader social, cultural and political factors which can affect the decision to vaccinate (e.g., travel bans or restrictions) [36]. It remains to be seen if these higher rates of vaccine uptake will be maintained over subsequent influenza seasons, or if the high rates observed this season were more of a 'one-off' reaction to being in the midst of the pandemic. However, importantly, despite some of the negative changes to vaccination attitudes currently observed, it is promising that we also identified high intentions to receive future doses of both seasonal influenza and COVID-19 vaccinations. As it is likely that booster doses of COVID-19 vaccinations will be required annually over the upcoming years [37], it is reassuring that this key population intends to continue to receive both vaccinations.

## Limitations

Our sample is not representative of all older adults in the UK. In particular, it comprised of participants who were living independently in the community and who were relatively high functioning. In addition, our sample was comprised almost entirely of white participants and may not represent the views and experience of more diverse racial groups. Minority ethnic groups in the UK have been reported to be more hesitant towards COVID-19 vaccinations and may experience inequities accessing COVID-19 vaccination services [38, 39]. By necessity, we also used online recruitment methods and an online survey to collect data for this study, which could have resulted in sampling bias. While using online surveys has been identified as a feasible way to collect data among older adults [40], factors such as limited internet access may hinder the generalizability of these findings. Finally, our vaccination uptake data is based on self-report and has not been verified by medical records. Self-reported vaccination status may overestimate vaccine uptake among older adults [41], and it is possible the widespread attention on COVID-19 vaccinations may have affected participants' responses.

## Conclusion

We identified positive shifts in vaccination attitudes during the first year of the COVID-19 pandemic, with complacency around vaccination, mistrust of vaccine benefit, concerns about commercial profiteering and constraints to vaccination all significantly decreasing over the year, and beliefs in the importance of collective responsibility significantly increasing. On the other hand, we also found that some attitudes to vaccination had become more negative, with participants now reporting being more concerned about potential vaccine side effects. In addition, we also found that participants would now engage in more deliberative information-seeking about vaccination. There is a need to ensure information-seeking older adults have access to credible information about vaccinations and potential side effects, along with disease risk, to inform their vaccination decisions.

## Supporting information

**S1 File. Survey used for data collection.**
(DOCX)

## Author Contributions

**Conceptualization:** Louise A. Brown Nicholls, Susan Rasmussen, Nicola Cogan, Lynn Williams.

**Data curation:** Allyson J. Gallant.

**Formal analysis:** Allyson J. Gallant, David Young.

**Funding acquisition:** Louise A. Brown Nicholls, Susan Rasmussen, Nicola Cogan, Lynn Williams.

**Investigation:** Allyson J. Gallant.

**Methodology:** Allyson J. Gallant, Louise A. Brown Nicholls, Susan Rasmussen, Nicola Cogan, Lynn Williams.

**Writing – original draft:** Allyson J. Gallant, Lynn Williams.

**Writing – review & editing:** Allyson J. Gallant, Louise A. Brown Nicholls, Susan Rasmussen, Nicola Cogan, David Young, Lynn Williams.

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
