## [Decision Letter · Decision Letter 0]

26 Jul 2021

PONE-D-21-21111

Changes in attitudes to vaccination as a result of the COVID-19 pandemic: A longitudinal study of older adults in the UK

PLOS ONE

Dear Dr. Williams,

Thank you for submitting your manuscript to PLOS ONE. After careful consideration, we feel that it has merit but does not fully meet PLOS ONE’s publication criteria as it currently stands. Therefore, we invite you to submit a revised version of the manuscript that addresses the points raised during the review process.

Two Reviewers have evaluated the manuscript, providing overall moderate revisions. I encourage Authors to submit a revised version, improving methodological details and Introduction/Discussion following Reviewers' suggestions.

We look forward to receiving your revised manuscript.

Kind regards,

Stefano Triberti, Ph.D.

Academic Editor

PLOS ONE

Journal Requirements:

[The study was funded by the Chief Scientist Office, Scottish Government (Ref: CGA/19/52).]

 [The study was funded by the Chief Scientist Office https://www.cso.scot.nhs.uk/, Scottish Government (Ref: Ref: CGA/19/52) research grant awarded to LW, LBN, SR, NC. The funders had no role in study design, data collection and analysis, decision to publish, or preparation of the manuscript.]

Reviewers' comments:

Reviewer's Responses to Questions

**Comments to the Author**

1. Is the manuscript technically sound, and do the data support the conclusions?

Reviewer #1: Yes

Reviewer #2: Yes

2. Has the statistical analysis been performed appropriately and rigorously? 

Reviewer #1: Yes

Reviewer #2: Yes

3. Have the authors made all data underlying the findings in their manuscript fully available?

Reviewer #1: No

Reviewer #2: Yes

4. Is the manuscript presented in an intelligible fashion and written in standard English?

Reviewer #1: Yes

Reviewer #2: Yes

5. Review Comments to the Author

Reviewer #1: Thanks for the opportunity to review this manuscript

In this study, the authors investigated an important and timely study question, which was related to the potential changes in attitudes of older adults (>65 years) in the UK towards vaccination using a longitudinal study design with two time points that were accompanied by Covid-19 pandemic.

The importance of such study question is the increased public discussion regarding vaccination in general and the widespread circulation of misinformation during Covid-19 pandemic which may have resulted in changes in public attitude towards vaccination (with possible decrease in confidence and increased calculation before getting vaccinated).

The strengths of this study involve the utilization of two separate time points to assess potential change in attitude towards flu vaccination among older adults and the utilization of previously validated scales assessing attitude towards vaccination (5C and VAX).

The results of the study pointed to calculation as the major psychologic determinant to be targeted among older adults in UK to tackle the problem of flu vaccine hesitancy.

In general, the article is well-written and the major limitations were addressed by the authors (especially in relation to representativeness of the study sample)

I have a few comments which might be considered by the authors to improve the manuscript as follows:

-The authors can benefit from additional relevant references in relation to Covid-19 vaccine hesitancy and its scope, flu and pneumococcal vaccine uptake among the elderly, among others

E.g. Betsch, C., Rossmann, C., Pletz, M.W. et al. Increasing influenza and pneumococcal vaccine uptake in the elderly: study protocol for the multi-methods prospective intervention study Vaccination60+. BMC Public Health 18, 885 (2018). https://doi.org/10.1186/s12889-018-5787-9

Schmid P, Rauber D, Betsch C, Lidolt G, Denker ML (2017) Barriers of Influenza Vaccination Intention and Behavior – A Systematic Review of Influenza Vaccine Hesitancy, 2005 – 2016. PLOS ONE 12(1): e0170550. https://doi.org/10.1371/journal.pone.0170550

Fridman A, Gershon R, Gneezy A (2021) COVID-19 and vaccine hesitancy: A longitudinal study. PLOS ONE 16(4): e0250123. https://doi.org/10.1371/journal.pone.0250123

-The authors could provide more details regarding the recruitment process (e.g. which social media platforms were used)

-The authors can benefit from mentioning the potential limitation involving sampling error that might have caused a biased estimation vaccine acceptance.

-Did the authors check for internal consistency of the 5C and VAX scales? If so, can they provide the Cronbach alpha values?

-Can the authors provide a decision or reference number of the ethical approval of the study by the School of Psychological Sciences & Health Ethics Committee at the University of Strathclyde?

-The authors are advised to submit the original questionnaire used in this study as a supplementary file to aid in the evaluation of their survey.

Reviewer #2: The paper is clear and it addresses a relevant topic. Nevertheless some major revisions are required. In particular,

Introduction

Please provide a brief description of the main tools used to investigate vaccine hesitancy (For example, see doi:10.15167/2421-4248/jpmh2020.61.3.1448)

Methods

- Recruiting partecipants through social media posts and using an online questionnaire could lead selection bias, especially if the study aims to investigate older adults' attitudes. Please discuss this issue.

Results

Table 1. What "M" means? Plaese clarify

Table 2. Please provide standard deviation/intrqurtile range

Discussion

There are some issues that could be adressed. In particular, the role of media and institutional comunication in determing attitudes towards vaccination should be described. Additionally it could be interesting discussing on the role of the public debate on vaccination policies. For example, possible restrictions (such travel ban) can determine intetion to get vaccinated?

6. PLOS authors have the option to publish the peer review history of their article (what does this mean?). If published, this will include your full peer review and any attached files.

Reviewer #1: **Yes: **Malik Sallam

Reviewer #2: No

---

## [Author Response · Author response to Decision Letter 0]

17 Aug 2021

Please see attached response to reviewers document.

---

## [Decision Letter · Decision Letter 1]

13 Dec 2021

Changes in attitudes to vaccination as a result of the COVID-19 pandemic: A longitudinal study of older adults in the UK

PONE-D-21-21111R1

Dear Dr. Williams,

We’re pleased to inform you that your manuscript has been judged scientifically suitable for publication and will be formally accepted for publication once it meets all outstanding technical requirements.

Kind regards,

Prof. Anat Gesser-Edelsburg, Ph.D.

Academic Editor

PLOS ONE

Additional Editor Comments (optional):

Reviewers' comments:

Reviewer's Responses to Questions

**Comments to the Author**

1. If the authors have adequately addressed your comments raised in a previous round of review and you feel that this manuscript is now acceptable for publication, you may indicate that here to bypass the “Comments to the Author” section, enter your conflict of interest statement in the “Confidential to Editor” section, and submit your "Accept" recommendation.

Reviewer #1: All comments have been addressed

2. Is the manuscript technically sound, and do the data support the conclusions?

Reviewer #1: Yes

3. Has the statistical analysis been performed appropriately and rigorously? 

Reviewer #1: Yes

4. Have the authors made all data underlying the findings in their manuscript fully available?

Reviewer #1: Yes

5. Is the manuscript presented in an intelligible fashion and written in standard English?

Reviewer #1: Yes

6. Review Comments to the Author

Reviewer #1: Thanks for fully addressing the points raised during the review process. Thus, I endorse the current manuscript for publication

7. PLOS authors have the option to publish the peer review history of their article (what does this mean?). If published, this will include your full peer review and any attached files.

Reviewer #1: **Yes: **Malik Sallam

---

## [Editor Report · Acceptance letter]

14 Dec 2021

PONE-D-21-21111R1 

Changes in attitudes to vaccination as a result of the COVID-19 pandemic: A longitudinal study of older adults in the UK 

Dear Dr. Williams:

I'm pleased to inform you that your manuscript has been deemed suitable for publication in PLOS ONE. Congratulations! Your manuscript is now with our production department. 

Kind regards, 

on behalf of

Prof. Anat Gesser-Edelsburg 

Academic Editor

PLOS ONE